# CAN WE GET THE BEST OF BOTH BINARY NEURAL NETWORKS AND SPIKING NEURAL NETWORKS FOR EFFICIENT COMPUTER VISION?

**Gourav Datta**[*,†]**, Zeyu Liu**[*]**, Peter A. Beerel**
University of Southern California, Los Angeles, CA, USA
{gdatta,liuzeyu,pabeerel}@usc.edu
[*]Equally contributing authors  [†]Currently employed at Amazon Inc.

## ABSTRACT

Binary Neural networks (BNN) have emerged as an attractive computing paradigm for a wide range of low-power vision tasks. However, state-of-the-art (SOTA) BNNs do not yield any sparsity, and induce a significant number of non-binary operations. On the other hand, activation sparsity can be provided by spiking neural networks (SNN), that too have gained significant traction in recent times. Thanks to this sparsity, SNNs when implemented on neuromorphic hardware, have the potential to be significantly more power-efficient compared to traditional artifical neural networks (ANN). However, SNNs incur multiple time steps to achieve close to SOTA accuracy. Ironically, this increases latency and energy—costs that SNNs were proposed to reduce—and presents itself as a major hurdle in realizing SNNs' theoretical gains in practice. This raises an intriguing question: *Can we obtain SNN-like sparsity and BNN-like accuracy and enjoy the energy-efficiency benefits of both?* To answer this question, in this paper, we present a training framework for sparse binary activation neural networks (BANN) using a novel variant of the Hoyer regularizer. We estimate the threshold of each BANN layer as the Hoyer extremum of a clipped version of its activation map, where the clipping value is trained using gradient descent with our Hoyer regularizer. This approach shifts the activation values away from the threshold, thereby mitigating the effect of noise that can otherwise degrade the BANN accuracy. Our approach outperforms existing BNNs, SNNs, and adder neural networks (that also avoid energy-expensive multiplication operations similar to BNNs and SNNs) in terms of the accuracy-FLOPs trade-off for complex image recognition tasks. Downstream experiments on object detection further demonstrate the efficacy of our approach. Lastly, we demonstrate the portability of our approach to SNNs with multiple time steps. Codes are publicly available here.

## 1 INTRODUCTION

Due to its low memory footprint and use of cheaper pop-count operations instead of energy-expensive multiply-and-accumulates (MAC), BNNs have emerged as a promising low-power alternative to compute- and memory-expensive deep neural networks (DNN) (Rastegari et al., 2016; Liu et al., 2020a; 2018a). Recent works have proposed novel network architectures (Bethge et al., 2021; Zhang et al., 2022; Shi et al., 2022; Hu et al., 2022) and training algorithms (Zhijun Tu & Wang, 2022a; Xu et al., 2021b; Rastegari et al., 2016; Bulat & Tzimiropoulos, 2019; Geng et al., 2023; Chen et al., 2021; Xu et al., 2021a; Kim et al., 2021; Wang et al., 2023; Lee et al., 2022) to approximate the full-precision representation of the weights and activations with 1-bit bi-polar values without significant drop in accuracy.

However, most of these efforts induce a significant number of non-binary operations which degrade the computational efficiency. For example, ReactNet-based BNNs (Liu et al., 2020a; Zhijun Tu & Wang, 2022b) incur custom non-linear functions, including RPReLU that are significantly more complex compared to threshold or ReLU operations, duplicated basic blocks that significantly increase the total number of floating point operations (FLOPs) and parameter count. Moreover, all

these BNNs employ the *sign* quantization function to generate the bi-polar weights and activation values. However, bi-polar activations do not yield any sparsity and thus can not benefit any computational efficiency from hardware that can leverage sparsity. Moreover, as shown in (Wang et al., 2020b; Lin et al., 2023; Falkena et al., 2023), it is also unclear whether *sign* is the optimal binarization function.

In contrast, SNNs can provide significant sparsity, even up to $80\%$ for ultra-low number of time steps (Chowdhury et al., 2021). However, this comes at the cost of significant accuracy drop (Datta et al., 2021) without any specialized neuron model (Datta et al., 2023b). In an attempt to push the frontier of sparsity-accuracy trade-off, we propose a class of uni-polar binary activation neural networks (BANNs) that can enjoy sparsity similar to SNNs while getting rid of the temporal dimension, and achieve BNN-like accuracies. Our BANNs are more compute-efficient than existing bi-polar BNNs. This efficiency stems from the fact that we can skip the memory access of the weight and the eventual accumulate/bit-count operation when the corresponding activation value is $0$.

**Our Contributions**. Our training framework for BANNs is based on a novel application of the Hoyer regularizer and a novel Hoyer thresholding layer. More specifically, our threshold is training-input-dependent and is set to be equal to the Hoyer extremum of a clipped version of the activation tensor, where the clipping value is trained using gradient descent with our Hoyer regularizer. In this way, compared to traditional uni-polar BNNs and SNNs with ultra-low number of time steps, our threshold increases the rate of weight updates and our Hoyer regularizer shifts the activation distribution away from this threshold, improving convergence. We consistently surpass the accuracies obtained by SOTA uni-polar BNNs (Wang et al., 2020b; Sakr et al., 2018) on diverse image recognition datasets with different convolutional architectures. Compared to SNNs, and adder neural network (AddNN) models that are also compute-efficient, our BANN models yield higher test accuracy with a $\sim5.5\times$ reduction in floating point operations (FLOPs), thanks to the extreme sparsity enabled by our training framework. Incorporating the training framework of BANNs can also improve the test accuracy in advanced bi-polar BNNs. Downstream tasks on object detection also demonstrate that our approach surpasses the test mAP of existing BNNs and SNNs. Our approach can also be incorporated to SNNs with multiple time steps, thereby leading to small but significant accuracy increase at the cost of significant increase in memory and compute cost. Thus, *our proposed approach acts as a continuum between one time-step sparse BNNs and low time-step SNNs, and can help bridge both the BNN and SNN communities for low-power vision tasks.*

## 2 PRELIMINARIES

### 2.1 HOYER REGULARIZER

Based on the interplay between $\ell_1$ and $\ell_2$ norms, a new measure of sparsity was first introduced in (Hoyer, 2004), based on which, (Yang et al., 2020) proposed a new regularizer, termed the Hoyer regularizer for the trainable weights that was incorporated into the loss term to train DNNs. We adopt the same form of Hoyer regularizer for the activation to train our BANN models as $H(\boldsymbol{u}_l) = \left(\frac{\|\boldsymbol{u}_l\|_1}{\|\boldsymbol{u}_l\|_2}\right)^2$ (Kurtz et al., 2020). Here, $\|\boldsymbol{u}_l\|_i$ represents the $\ell_i$ norm of the activation tensor $\boldsymbol{u}_l$, and the superscript $t$ for the time step is omitted for simplicity. Compared to the $\ell_1$ and $\ell_2$ regularizers, the Hoyer regularizer has scale-invariance (similar to the $\ell_0$ regularizer). It is also differentiable almost everywhere (see equation 1) where $|\boldsymbol{u}_l|$ represents the element-wise absolute of the tensor $\boldsymbol{u}_l$.

$$\frac{\partial H(\boldsymbol{u}_l)}{\partial \boldsymbol{u}_l} = \frac{2\|\boldsymbol{u}_l\|_1}{\|\boldsymbol{u}_l\|_2^2}\left(sign(\boldsymbol{u}_l)\cdot\|\boldsymbol{u}_l\|_2 - \frac{\|\boldsymbol{u}_l\|_1}{\|\boldsymbol{u}_l\|_2}\boldsymbol{u}_l\right) \tag{1}$$

Letting the gradient $\frac{\partial H(\boldsymbol{u}_l)}{\partial u_l}=0$ and making all the $\boldsymbol{u}_l$ positive, the value of the Hoyer extremum becomes $E(\boldsymbol{u}_l)=\frac{\|\boldsymbol{u}_l\|_2^2}{\|\boldsymbol{u}_l\|_1}$. This extremum is the minimum, because the second derivative is greater than zero for any value of the output element. Training with the Hoyer regularizer can effectively help push the activation values that are larger than the extremum ($\boldsymbol{u}_l>E(\boldsymbol{u}_l)$) even larger and those that are smaller than the extremum ($\boldsymbol{u}_l<E(\boldsymbol{u}_l)$) even smaller.

## 2.2 BINARY NEURAL NETWORKS

Given a input $o_{l-1} \in \mathbb{R}^{c \times h \times w}$ and weight $w \in \mathbb{R}^{n \times c \times k \times k}$ of a layer $l$, we can get the output $u_l \in \mathbb{R}^{c \times h' \times w'}$ by convolution operation as $u_l = o_{l-1} * w_l$. To accelerate the inference process, bi-polar BNNs (Liu et al., 2020a; Zhijun Tu & Wang, 2022b) partition the input and weight into two clusters, -1 and +1 with sign function as Eq. 2.

$$sign(x) = \begin{cases} 1, & \text{if } x \geq 0; \\ -1, & \text{otherwise} \end{cases} \tag{2}$$

To yield sparsity similar to ReLU activation, uni-polar BNNs (Wang et al., 2020b; Sakr et al., 2018) parition the weights to -1 and +1, while the activations to 0 and 1 as illustrated below in Eq. 3.

$$z_l = \frac{u_l}{v_l^{th}} \qquad o_l = \begin{cases} 1, & \text{if } z_l \geq 1; \\ 0, & \text{otherwise} \end{cases} \tag{3}$$

where $z_l$ denotes the normalized activation output, and $v_l^{th}$ denotes a trainable threshold parameter. While our BANNs primarily focus on sparse uni-polar activations with full-precision weights (activations are significantly more difficult to binarize compared to weights (Lin et al., 2017b; Ding et al., 2019)), we also demonstrate the efficacy of BANNs with binary weights (-1 and +1) for a fair comparison with BNNs.

## 3 PROPOSED TRAINING FRAMEWORK

Our approach is inspired by the fact that Hoyer regularizers can shift the pre-activation distributions away from the Hoyer extremum in a DNN (Yang et al., 2020). Our principal insight is that setting our activation threshold to this extremum shifts the distribution away from the threshold value, reducing noise and improving convergence. To obtain our BANNs, we present a novel *Hoyer threshold layer* that sets the threshold based upon a *Hoyer regularized training process*, as described below.

### 3.1 HOYER THRESHOLD LAYER

As illustrated above in Eq. 3, a binary neuron with a unit step activation function is difficult to optimize with straight through estimator (STE) based approaches (Bengio et al., 2013), which either approximates the binary neuron functionality with a continuous differentiable model or surrogate gradient approaches used in SNNs (Panda & Roy, 2016; Lee et al., 2016). This is because the number of *ones* in the activations becomes too low to adjust the weights sufficiently using

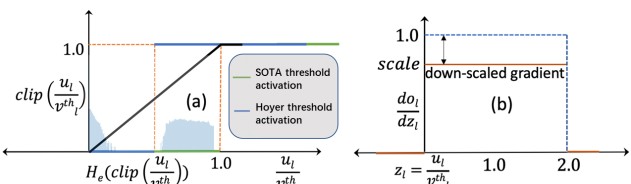

Figure 1: (a) Comparison of our Hoyer threshold function with existing uni-polar binary activation functions where the blue distribution denotes the shifting of pre-activation values away from the threshold using Hoyer regularized training, (b) Proposed derivative of our Hoyer threshold function.

gradient descent. If a pre-synaptic neuron does not output 1, the synaptic weight connected to it cannot be updated because its gradient from neuron $i$ is calculated as $g_{u_j} \times o_i$, where $g_{u_j}$ is the gradient of the activation $u_j$ and $o_i$ is the output of neuron $i$.

Therefore, it is crucial to reduce the value of the threshold to generate enough 1s for better network convergence. However, a sufficiently low value of threshold can generate a *one* for every neuron, but that would yield random outputs in the final classifier layer. Hence, it is challenging to yield the optimal threshold for each layer. Previous works (Wang et al., 2020b) show that this problem can be partially mitigating by training the threshold term $v_l^{th}$ using gradient descent. However, that still leads to a large accuracy gap with the SOTA bi-polar BNNs, especially for ImageNet-level tasks.

In contrast, we propose to dynamically down-scale the threshold (see Fig. 1(a)) based on the activation using our proposed form of the Hoyer regularizer. In particular, we clip the activation

corresponding to each convolutional layer to the trainable threshold $v_l^{th}$ obtained from the gradient descent with our Hoyer loss, as detailed later in Eq. 12. Unlike existing approaches (Rathi et al., 2020a; Chowdhury et al., 2021) that require $v_l^{th}$ to be initialized from a pre-trained full-precision model, our approach can be used to train BANNs from scratch with a Kaiming uniform initialization (He et al., 2015) for both the weights and thresholds. In particular, the normalized down-scaled threshold value, with which we compare the normalized activation $z_l$, for each layer is computed as the Hoyer extremum of the clipped activation tensor as shown in Fig. 1(a) and below.

$$
\boldsymbol{z}_l^{clip} = \begin{cases} 1, \text{if } \boldsymbol{z}_l > 1 \\ \boldsymbol{z}_l, \text{if } 0 \leq \boldsymbol{z}_l \leq 1 \\ 0, \text{if } \boldsymbol{z}_l < 0 \end{cases} \quad \boldsymbol{o}_l = h_s(\boldsymbol{z}_l) = \begin{cases} 1, \text{if } \boldsymbol{z}_l \geq E(\boldsymbol{z}_l^{clip}) \\ 0, \text{otherwise} \end{cases} \tag{4}
$$

Note that our normalized threshold $E(\boldsymbol{z}_l^{clip})$ is less than the normalized threshold whose value is 1 for any output (proof in supplementary materials). Hence, our actual threshold value $E(\boldsymbol{z}_l^{clip}) \times v_l^{th}$ is indeed less than the trainable threshold $v_l^{th}$ used in earlier works (Datta & Beerel, 2022; Rathi et al., 2020a). We also observe that the Hoyer extremum in each layer changes only slightly during the later stages of training, which indicates that it is most likely an inherent attribute of the dataset and model architecture. Hence, to estimate the threshold during inference, we calculate the exponential average of the Hoyer extremums during training, similar to batch normalization (BN) layers, and use the same during inference.

## 3.2 HOYER REGULARIZED TRAINING

The loss function ($L_{total}$) of our proposed approach is shown below in Eq. 5.

$$
L_{total} = L_{CE} + \lambda_H L_H = L_{CE} + \lambda_H \sum_{l=1}^{L-1} H(\boldsymbol{z}_l^{clip}) \tag{5}
$$

where $L_{CE}$ denotes the cross-entropy loss calculated on the softmax output of the last layer $L$, and $L_H$ represents the Hoyer regularizer calculated on the input of our Hoyer threshold layer after dividing the threshold term $v_l^{th}$ and clipping. The weight update for the layer $L-1$ is computed as

$$
\begin{aligned}
\Delta W_{L-1} &= \frac{\partial L_{CE}}{\partial \boldsymbol{w}_{L-1}} + \lambda_H \frac{\partial L_H}{\partial \boldsymbol{w}_{L-1}} = \frac{\partial L_{CE}}{\partial \boldsymbol{o}_{L-1}} \frac{\partial \boldsymbol{o}_{L-1}}{\partial \boldsymbol{u}_{L-1}} \frac{\partial \boldsymbol{u}_{L-1}}{\partial \boldsymbol{w}_{L-1}} + \lambda_H \frac{\partial L_H}{\partial \boldsymbol{u}_{L-1}} \frac{\partial \boldsymbol{u}_{L-1}}{\partial \boldsymbol{w}_{L-1}} \\
&= \left( \frac{\partial L_{CE}}{\partial \boldsymbol{o}_{L-1}} \frac{\partial \boldsymbol{o}_{L-1}}{\partial \boldsymbol{u}_{L-1}} + \lambda_H \frac{\partial H(\boldsymbol{z}_{L-1}^{clip})}{\partial \boldsymbol{u}_{L-1}} \right) \boldsymbol{o}_{L-2}
\end{aligned} \tag{6}
$$

$$
\frac{\partial L_{CE}}{\partial \boldsymbol{o}_{L-1}} = \frac{\partial L_{CE}}{\partial \boldsymbol{u}_L} \frac{\partial \boldsymbol{u}_L}{\partial \boldsymbol{o}_{L-1}} = (\boldsymbol{s} - \boldsymbol{y}) \boldsymbol{w}_L \tag{7}
$$

where $\boldsymbol{s}$ denotes the output softmax tensor, i.e., $s_i = \frac{e^{u_L^i}}{\sum_{k=1}^N u_L^k}$ where $u_L^i$ and $u_L^k$ denote the $i^{th}$ and $k^{th}$ elements of the activation tensor of the last layer $L$, and $N$ denotes the number of classes. Note that $\boldsymbol{y}$ denotes the one-hot encoded tensor of the true label, and $\frac{\partial H(\boldsymbol{u}_L)}{\partial \boldsymbol{u}_L}$ is computed using Eq. 1. The last layer does not have any threshold and yields full-precision outputs.

The weight and threshold update computations for the hidden layers are in Appendix A.1.

Note that all the derivatives to update the trainable parameters can be computed by Pytorch autograd, except the derivative $\frac{\partial \boldsymbol{o}_l}{\partial \boldsymbol{z}_l}$, whose gradient is zero almost everywhere and undefined at $z_l = 0$. We extend the existing idea of surrogate gradient descent (SGD) (Neftci et al., 2019) in SNNs to compute this derivative for our BANNs with Hoyer threshold layers, as illustrated in Fig. 1(b) and mathematically defined as follows.

$$
\frac{\partial \boldsymbol{o}_l}{\partial \boldsymbol{z}_l} = \begin{cases} scale \times 1 & \text{if } 0 < \boldsymbol{z}_l < 2 \\ 0 & \text{otherwise} \end{cases} \tag{8}
$$

where *scale* denotes a hyperparameter that controls the dampening of the gradient. Note that $o_l$ jumps from 0 to 1 at $z_l = 1$. We assume the surrogate gradient is non-zero spanning a width of 1 in both directions around $z_l = 1$.

Table 1: Comparison of the test accuracy of our BANNs with full-precision DNNs for object recognition. Model* indicates that we remove the first max pooling layer, and Sp. denotes sparsity.

| Network | dataset | DNN (%) | BANN (%) | Sp. (%) |
|---------|---------|---------|----------|---------|
| VGG16 | CIFAR10 | 94.10 | 93.44 | 78.13 |
| ResNet18 | CIFAR10 | 93.34 | 92.50 | 74.17 |
| ResNet18* | CIFAR10 | 94.28 | 93.88 | 83.88 |
| ResNet20 | CIFAR10 | 93.18 | 92.39 | 76.31 |
| ResNet34* | CIFAR10 | 94.68 | 93.53 | 83.96 |
| ResNet50* | CIFAR10 | 94.90 | 94.07 | 82.21 |
| VGG16 | ImageNet | 70.08 | 68.00 | 75.52 |
| ResNet50 | ImageNet | 73.12 | 66.98 | 76.11 |

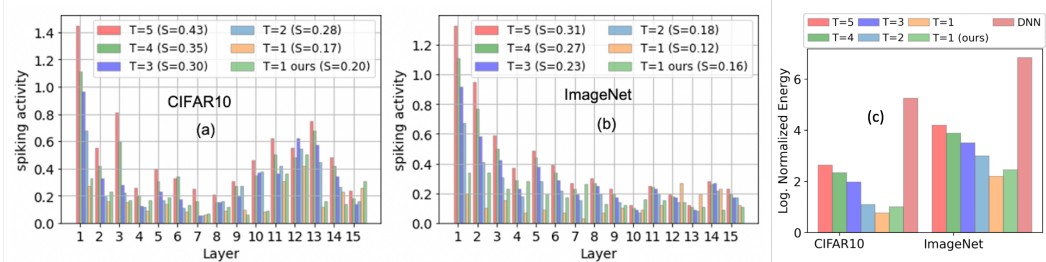

Figure 2: Layerwise spiking activities for a VGG16 across time steps ranging from 5 to 1 (average spiking activity denoted as $S$ in parenthesis) representing existing low-latency SNNs including our work (which represents one time step) on (a) CIFAR10, (b) ImageNet, (c) Comparison of the total energy consumption between SNNs with different time steps and full-precision DNNs.

## 4 EXPERIMENTAL RESULTS

**Datasets & Models**: Similar to existing BNN and SNN works (Rathi et al., 2020b;a; Liu et al., 2020a), we perform object recognition experiments on CIFAR10 (Krizhevsky, 2009) and ImageNet (Deng et al., 2009) dataset using VGG16 (Simonyan & Zisserman, 2014) and several variants of ResNet (He et al., 2016) architectures. For object detection, we use the MMDetection framework (Chen et al., 2019) with PASCAL VOC2007 and VOC2014 (Everingham et al., 2010) as training dataset, and benchmark our BANN models and the baselines on the VOC2007 test dataset. We use the Faster R-CNN (Ren et al., 2015) and RetinaNet (Lin et al., 2017a) framework, and substitute the original backbone with our BANN models pretrained on ImageNet.

**Object Recognition Results**: For training the recognition models, we use the Adam (Kingma & Ba, 2014) optimizer for VGG16, and use SGD optimizer for ResNet models. As shown in Table 1 with FP weights, we obtain the SOTA accuracy of $93.44\%$ on CIFAR10 with VGG16; the accuracy of our ResNet-based BANNs on ImageNet also surpasses the existing works. Note that our ResNet models are based on the spike-element-wise (SEW) architecture (Fang et al., 2020) as detailed in Appendix A.2. On ImageNet, we obtain a $68.00\%$ top-1 accuracy with VGG16 which is only $\sim 2\%$ lower compared to the iso-architecture full-precision counterpart. All our BANN models yield a sparsity of $\sim 75\%$ or higher on both CIFAR10 and ImageNet, which is significantly higher compared to existing SNNs/BNNs as shown in Fig. 2.

**Accuracy Comparison with SNNs**: We compare our results with various SOTA ultra low-latency SNNs for image recognition tasks in Table 2. Our BANNs yield comparable or better test accuracy compared to all the existing works on both CIFAR10 and ImageNet, with significantly lower inference latency (due to the existence of multiple time steps in SOTA SNNs). The only exception for the latency reduction is the one-time-step SNN proposed in (Chowdhury et al., 2021), however, it increases the training time significantly as illustrated later in Fig. 3.

**Energy Efficiency Comparison with SNNs and full-precision DNNs**: We compare the energy-efficiency of our BANNs with full-precision DNNs and existing multi-time-step SNNs in Fig. 2. The compute-efficiency of BANNs stems from two factors: 1) sparsity, that reduces the number of synaptic operations in convolutional and linear layers compared to full-precision DNNs according to $\text{SNN}_l^{flops} = S_l \times \text{DNN}_l^{flops}$ (Chowdhury et al., 2021), where $S_l$ denotes the average number of spikes per neuron per inference over all timesteps in layer $l$. Note that the sparsity induces a small

Table 2: Comparison of our BANN models to existing SNNs. SGD and hybrid denote surrogate gradient descent and pre-trained DNN followed by SNN fine-tuning respectively. (qC, dL) denotes an architecture with q convolutional and d linear layers. T.S denotes the number of time steps. *Results reported without autoaugmentation and cutout for fair comparison with our work.

| Ref. | Training | Architecture | Acc. (%) | T.S |
|---|---|---|---|---|
| Dataset : CIFAR10 | | | | |
| (Meng et al., 2022) | SGD | PreAct-ResNet19 | **95.40** | 20 |
| (Deng et al., 2021) | DNN-SNN conversion | VGG16 | 92.29 | 16 |
| (Bu et al., 2022a) | DNN-SNN coonversion | VGG16 | 90.96 | 8 |
| (Zhang & Li, 2020) | SGD | 5C, 2L | 91.41 | 5 |
| (Rathi et al., 2020a) | Hybrid | VGG16 | 92.70 | 5 |
| (Zheng et al., 2021) | STBP-tdBN | ResNet19 | 93.16 | 6 |
| (Datta & Beerel, 2022) | Hybrid | VGG16 | 91.79 | 2 |
| (Bu et al., 2022b) | DNN-SNN conversion | VGG16 | 91.18 | 2 |
| (Fang et al., 2020) | SGD | 5C, 2L | 93.50 | 8 |
| (Deng et al., 2023)* | SGD | ResNet18 | **94.58** | 2 |
| (Xiao et al., 2022) | SGD | VGG (sWS) | 93.73 | 6 |
| (Deng et al., 2022) | SGD | ResNet19 | 94.16 | 2 |
| (Li et al., 2021b) | SGD | ResNet18 | 93.13 | 2 |
| (Chowdhury et al., 2021) | Hybrid | VGG16 | 93.05 | 1 |
| (Chowdhury et al., 2021) | Hybrid | ResNet20 | 91.10 | 1 |
| **Ours** | **Adam+Hoyer Reg.** | **VGG16** | **93.44** | **1** |
| Dataset : ImageNet | | | | |
| (Bu et al., 2022b) | DNN-SNN conversion | ResNet34 | 59.35 | 16 |
| (Li et al., 2021b) | SGD | ResNet18 | **71.24** | 5 |
| (Rathi et al., 2020a) | Hybrid | VGG16 | 69.00 | 5 |
| (Fang et al., 2021) | SGD | ResNet34 | 67.04 | 4 |
| (Fang et al., 2021) | SGD | ResNet152 | **69.26** | 4 |
| (Deng et al., 2023)* | SGD | ResNet34 | 65.77 | 2 |
| (Duan et al., 2022) | SGD | ResNet34 | 68.28 | 4 |
| (Deng et al., 2022) | SGD | ResNet19 | 68.00 | 4 |
| (Rathi et al., 2020a) | Hybrid | VGG16 | 69.00 | 5 |
| (Zheng et al., 2021) | STBP-tdBN | ResNet34 | 67.05 | 6 |
| (Chowdhury et al., 2021) | Hybrid | VGG16 | 67.71 | 1 |
| **Ours** | **Adam+Hoyer Reg.** | **VGG16** | **68.00** | **1** |

overhead of checking whether the 1-bit activation is zero, which consumes 0.05pJ in 28nm Kintex-7 FPGA platform according to our post place-and-route simulations. 2) Use of only AC (1.8pJ) operations that consume $7.4\times$ lower compared to each MAC (13.32pJ) operation in our FPGA setup for floating point (FP) representation. Note that the binary activations can replace the FP multiplications with logical operations, i.e., conditional assignment to 0 with a bank of AND gates. These replacements may be realized using existing hardware depending on the compiler and the details of their data paths. Building a custom accelerator that can efficiently implement these reduced operations is also possible (Wang et al., 2020a; Frenkel et al., 2019; Lee & Li, 2020). Additionally, our BANNs also enjoy superior memory-efficiency compared to existing multi-time-step SNNs since the latter requires the membrane potentials and weights to be fetched from and read to the on-/off-chip memory for each time step. Our BANNs can avoid these repetitive read/write operations as it does involve any *state* and lead to a $T\times$ reduction in the number of memory accesses compared to a T-time-step SNN model. Compared to traditional bi-polar BNNs, our BANNs can also reduce the number of memory accesses with the support of zero gating logic leveraging the high activation sparsity. This can be achieved by skipping the reading of the weights when the activation associated to it is zero. However, the exact savings in memory energy will depend on the weight reuse scheme and the underlying hardware. We have provided the detailed energy model of traditional BNNs, multi-time-step SNNs, BANNs, and full-precision DNNs, considering the sparsity overhead and memory accesses without any weight reuse scheme, in Appendix A.9. We compare the layer-wise spiking activities $S_l$ for time steps ranging from 5 to 1 in Fig. 2(a-b) that are computed based on the

Table 3: Accuracies from different strategies to train BANNs on CIFAR10

| Training Strategies | Pretrained DNN(%) | BANN (%) | Sparsity (%) |
|---|---|---|---|
| Pre-trained+fine-tuning | 93.15 | 91.39 | 23.56 |
| Iterative training (N=10) | 93.25 | 92.68 | 10.22 |
| Iterative Training (N=20) | 92.68 | 92.24 | 9.54 |
| Proposed Training | - | **93.13** | 22.57 |

Table 4: Accuracies and energies of weight quantized BANN models based on VGG16 on CIFAR10 where FP is 32-bit floating point.

| Dataset | Bits | Acc. (%) | Spiking Activity (%) | Energy (mJ) |
|---|---|---|---|---|
| CIFAR10 | FP | 93.44 | 22.57 | 18.67 |
| CIFAR10 | 6 | 93.11 | 22.46 | 7.46 |
| CIFAR10 | 2 | 92.94 | 21.39 | 5.66 |
| CIFAR10 | 1 | 92.80 | 22.68 | 5.02 |
| ImageNet | FP | 68.00 | 24.48 | 424.3 |
| ImageNet | 2 | 66.86 | 23.82 | 231.3 |
| ImageNet | 1 | 64.62 | 23.99 | 221.8 |

replicated implementation of an existing low-latency SNN (Rathi et al., 2020a), against our work (where the spiking activity denotes the number of 1s) that represents a time step of 1. Note, the spike rates decrease significantly with time step reduction from 5 to 1, leading to considerably lower FLOPs in our one-time-step SNNs. These lower FLOPs, coupled with the $7.4\times$ reduction for AC operations and the sparsity overhead, lead to a $22.7\times$ and $31.9\times$ reduction in compute energy on CIFAR10 and ImageNet respectively with VGG16. Considering the memory accesses as provided in our energy models in A.9, the total energy reduces by $70.28\times$ and $81.13\times$.

**Comparison with different training strategies**: Based on existing BNN and SNN literature, we hypothesize that two training strategies that can be effectively used to train BANNs, other than our proposed approach.

*Pre-trained DNN, followed by BANN fine-tuning.* Similar to the hybrid training proposed in (Rathi et al., 2020b), we pre-train a full-precision DNN model, and copy its weights to the BANN model. Initialized with these weights, we train our BANN with normal cross-entropy loss.

*Iteratively convert ReLU neurons to binary neurons.* First, we train a DNN model which uses the ReLU activation with threshold, then we iteratively reduce the number of the ReLU neurons whose output activations are multi-bit. Specifically, we first force the activation values in the top $N$ percentile to output 1, and those in bottom $N$ percentile percent to output 0, and gradually increase $N$ until the accuracy drops beyond a certain threshold or all neuron outputs are either 1 or 0.

*Proposed training from scratch.* With our proposed Hoyer threshold layer and Hoyer regularized training, we train a BANN model from scratch. Our results with these training strategies are shown in Table 3, which indicates that it is difficult for training strategies that involve pre-training and fine-tuning to approach the accuracy of full-precision models with BANNs. One possible reason for this might be the difference in the distribution of the pre-activation values between the DNN and BANN models (Datta & Beerel, 2022). Our Hoyer threshold layer and our Hoyer regularizer, can help train a BANN model with SOTA accuracy from scratch.

**Effect on Quantization**: In order to further improve the energy efficiency of our BANNs, we perform quantization-aware training of the weights in our models to $1-6$ bits for both CIFAR10 and ImageNet. This transforms the FP ACs to $1-6$ bit ACs, thereby leading to a $4.8-13.8$ reduction in compute energy as obtained from our FPGA simulations. The reduced weight precision also reduces the memory access cost by $\sim 5-32\times$ with efficient bit-packing (bit, 2021) for $1-6$ bits according to our FPGA simulations. Note that we only quantize the convolutional layers, as quantizing the linear layers lead to a noticeable drop in accuracy. As shown in Table 4, when quantized to 6 bits, our VGG-based BANN incur a negligible accuracy drop of only $0.02\%$ on CIFAR10, while reducing the total energy by $3.3\times$ (energy computed from the models developed in A.9). Even with 1-bit quantization, our model can yield an accuracy of $92.80\%$ without any special modification, while still yielding a sparsity of $\sim 78\%$.

**Comparison with AddNNs & BNNs**: We compare the accuracy and energy of our BANN

Table 5: Accuracy & Energy Comparison of our BANNs to AddNNs and BNNs

| Reference | Dataset | Acc.(%) | Energy (mJ) |
|---|---|---|---|
| Uni-polar BNNs | | | |
| (Sakr et al., 2018) | CIFAR10 | 89.6 | 4.94 |
| (Wang et al., 2020b) | ImageNet | 59.7 | 215.0 |
| Bi-polar BNNs | | | |
| (Diffenderfer & Kailkhura, 2021) | CIFAR10 | 91.9 | 14.72 |
| AddNNs | | | |
| (Chen et al., 2020) (FP weights) | CIFAR10 | 93.72 | 95.22 |
| (Chen et al., 2020) (2-bit weights) | CIFAR10 | 92.08 | 24.57 |
| (Chen et al., 2020) (FP weights) | ImageNet | 67.0 | 2291.2 |
| Our BANNs | | | |
| This work (FP weights) | CIFAR10 | 93.44 | 18.67 |
| This work (1-bit weights) | CIFAR10 | 92.80 | 5.02 |
| This work (FP weights) | ImageNet | 68.00 | 424.3 |
| This work (1-bit weights) | ImageNet | 64.62 | 221.8 |

Table 7: Test accuracy obtained by our approach with multiple time steps for SNNs on CIFAR10.

| Architecture | Time steps | Acc. (%) | Spiking activity (%) |
|---|---|---|---|
| VGG16 | 1 | 93.44 | 21.87 |
| VGG16 | 2 | 93.71 | 44.06 |
| VGG16 | 6 | 94.14 | 101.22 |
| ResNet18 | 1 | 91.48 | 25.83 |
| RseNet18 | 2 | 91.93 | 33.24 |
| ResNet18 | 6 | 92.01 | 83.82 |

models with recently proposed AddNN models (Chen et al., 2020) that also removes multiplications for increased energy-efficiency in Table 5. With the VGG16 architecture, on CIFAR10, we obtain $0.6\%$ lower accuracy, while on ImageNet, we obtain $1.0\%$ higher accuracy. Moreover, unlike our BANNs and SNNs, AddNNs do not yield any sparsity, and consume $\sim 5.1 \times$ more compute energy (with the sparsity overhead incorporated) compared to our BANNs on average across both CIFAR10 and ImageNet (see Table 5). We also compare our BANN models with SOTA BNNs in Table 7 that replaces the costly MAC operations with cheaper pop-count counterparts, thanks to the binary weights and activations. Our BANNs with 2-bit quantized weights consume $2.6 \times$ lower energy compared to the bi-polar BNNs (see (Diffenderfer & Kailkhura, 2021) in Table 5). This is due to the improved trade-off between the high sparsity activity ($\sim 78\%$ as shown in Table 1)

Table 6: Impact of our proposed Hoyer regularized training framework on the test accuracy with advanced BNNs

| Work | Acc. (%) | Acc. w/ Hoyer (%) |
|---|---|---|
| PokeBNN-1.0x | 73.4 | 73.6 (+0.2) |
| MeliusNet59 | 71.0 | 72.3 (+1.3) |
| ReCU (Resnet18) | 66.4 | 67.8 (+1.4) |

provided by our BANN models, and less energy due to XOR operations and 1-bit memory access compared to weight-quantized ACs and 2-bit memory access. On the other hand, our BANNs with 1-bit weights i.e., our sparse uni-polar BNNs consume similar energy compared to existing unipolar BNNs (see (Sakr et al., 2018; Wang et al., 2020b) in Table 5) while yielding $2.6\%$ higher accuracy ($92.80\%$ vs $90.2\%$) on CIFAR10 and $4.9\%$ higher accuracy ($64.62\%$ vs $59.7\%$) on ImageNet at iso-architecture. Lastly, it is worth mentioning that though our sparse BNNs can surpass test accuracies obtained by a few existing BNNs, it fails to yield accuracies similar to the SOTA bi-polar BNNs (Zhang et al., 2022; Bethge et al., 2021; Zhijun Tu & Wang, 2022a; Xu et al., 2021b). This is due to their higher expressivity (further explained in A.11) and network modifications (e.g. ReactNet with PReLU compared to our simple VGG and ResNets) that increases the FLOPs by more than $2 \times$. However, applying our proposed training technique on these BNNs results in

$0.2-1.3\%$ increase in test accuracy as shown in Table 6. This is done using the same way as using the Hoyer extremum of the normalized and clipped activation as the threshold, where the clipping is done at -1 and +1 (instead of 0 and +1 for our BANNs). This demonstrate the efficacy of our proposed method.

**Extension to SNNs**: We extend our proposed approach to multi-time-step SNN models by adopting the standard LIF model (Sengupta et al., 2019) for the neurons but computing the threshold based on the Hoyer extremum defined in Eq. 3 and unrolling the gradients derived in Section 3.2 using traditional backpropagation through time (Rathi et al., 2020a). As shown in Table 7, as time step increases from 1 to 6, the accuracy of our model also increases from 93.44% to 94.14%, which validates the effectiveness of our method. However, this accuracy increases comes at the cost of a significant increase in spiking activity (see Table 7 where the spiking activity is computed across the total number of time steps similar to (Sengupta et al., 2019)), increasing the compute energy. The memory cost also increases due to the repetitive potential and weight accesses across time steps.

## 5    RELATIONSHIP BETWEEN SNN, BNN, AND BANN

Our BANNs are identical to uni-polar sparse BNNs (Sakr et al., 2018; Wang et al., 2020b) when the weight precision is quantized down to 1-bit. Our BANNs are also identical to one-time-step SNNs, and can be readily extended to traditional multi-time-step SNNs with the incorporation of the membrane potential, using the traditional LIF model for the neurons where the membrane potential integrates the weight modulated input spikes and leaks over time. We use the soft reset mechanism that reduces the membrane potential by the threshold value when an output spike generated (Datta et al., 2023a). It has been shown that soft reset minimizes the information loss by allowing the spiking neuron to carry forward the surplus potential above the firing threshold to the subsequent time step (). We use our proposed combination of Hoyer regularized training and Hoyer spike layer to train the per layer threshold, while we train the weights and leak term using SGL. The dynamics of our LIF model for our multi-time-step SNNs are shown below.

$$U_i^{mp}(t) = \lambda U_i(t-1) + \sum_j W_{ij} S_j(t), \quad S_i(t) = 1 \text{ if } U_i^{mp}(t) > V^{th} \text{ else } 0, \quad U_i(t) = U_i^{mp}(t) - S_i(t) V^{th}$$

Here $U_i(t)$ denotes the membrane potential of the $i^{th}$ neuron at timer step $t$, $S_i(t)$ denotes the binary output of the $i^{th}$ neuron at time step $t$, $V^{th}$ denotes the threshold, $\lambda$ denotes the leak, and $W_{ij}$ denotes the weight connecting the pre-synaptic neuron $j$ and the neuron $i$.

The sparse activation in BANNs bears resemblance to SNNs while the difference is that there is no temporal dimension. On the other hand, the 1-bit sparse activation in BANNs bears resemblance to BNNs, in particular uni-polar BNNs, while the difference is that the weight precision is multi-bit. Thus, BANNs can bridge the gap between BNNs and SNNs by bringing the activation sparsity of SNNs to BNNs, and removing the temporal dimension of SNNs for equivalence with BNNs.

## 6    CONCLUSIONS

In this work, we propose BANNs trained using a novel variant of Hoyer Regularizer and a Hoyer threshold layer that jointly optimizes the distribution of the BANN pre-activations and the placement of the BANN threshold in order to improve the accuracy-FLOPs trade-off compared to existing networks. Our Hoyer regularizer introduces a loss term that penalizes the normalized and clipped activation values around their Hoyer extremum (which depends on the SGD trained threshold). Our proposed Hoyer threshold layer takes advantage of this distribution shift by setting the IF threshold to the Hoyer extremum. Our BANN models surpass test accuracies obtained by existing SNN, uni-polar BNN, and AddNN models with similar or better energy-efficiency. Our training framework can also improve the accuracy of a number of advanced bi-polar BNNs. Our SNNs can further improve the inference energy and latency efficiency with recently proposed in-sensor computing systems (Datta et al., 2022d;a;c) where the bandwidth between the sensor that implements the first few CNN layers and the back-end processing unit that implements the remaining layers can be significantly reduced, thanks to the binary activations for only one time step (Datta et al., 2023c).

## 7    ACKNOWLEDGEMENT

This work is supported by a gift funding from Intel Labs.

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

# A APPENDIX

## A.1 THRESHOLD & WEIGHT UPDATES DUE TO HOYER LOSS

For a hidden layer $l$, the weight update is computed as

$$
\begin{aligned}
\Delta W_l &= \frac{\partial L_{CE}}{\partial \boldsymbol{w}_l} + \lambda_H \frac{\partial L_H}{\partial \boldsymbol{w}_l} = \frac{\partial L_{CE}}{\partial \boldsymbol{o}_l} \frac{\partial \boldsymbol{o}_l}{\partial \boldsymbol{z}_l} \frac{\partial \boldsymbol{z}_l}{\partial \boldsymbol{u}_l} \frac{\partial \boldsymbol{u}_l}{\partial \boldsymbol{w}_l} \\
&+ \lambda_H \frac{\partial L_H}{\partial \boldsymbol{u}_l} \frac{\partial \boldsymbol{u}_l}{\partial \boldsymbol{w}_l} = \frac{\partial L_{CE}}{\partial \boldsymbol{o}_l} \frac{\partial \boldsymbol{o}_l}{\partial \boldsymbol{z}_l} \frac{\boldsymbol{o}_{l-1}}{v_l^{th}} + \lambda_H \frac{\partial L_H}{\partial \boldsymbol{u}_l} \boldsymbol{o}_{l-1}
\end{aligned}
\tag{9}
$$

where $\frac{\partial L_H}{\partial \boldsymbol{u}_l}$ can be computed as

$$
\begin{aligned}
\frac{\partial L_H}{\partial \boldsymbol{u}_l} &= \frac{\partial L_H}{\partial \boldsymbol{u}_{l+1}} \frac{\partial \boldsymbol{u}_{l+1}}{\partial \boldsymbol{o}_l} \frac{\partial \boldsymbol{o}_l}{\partial \boldsymbol{z}_l} \frac{\partial \boldsymbol{z}_l}{\partial \boldsymbol{u}_l} + \frac{\partial H(\boldsymbol{z}_l^{clip})}{\partial \boldsymbol{u}_l} \\
&= \frac{\partial L_H}{\partial \boldsymbol{u}_{l+1}} \boldsymbol{w}_{l+1} \frac{\partial \boldsymbol{o}_l}{\partial \boldsymbol{z}_l} \frac{1}{v_l^{th}} + \frac{\partial H(\boldsymbol{z}_l^{clip})}{\partial \boldsymbol{z}_l^{clip}} \frac{\partial \boldsymbol{z}_l^{clip}}{\partial \boldsymbol{z}_l} \frac{1}{v_l^{th}}
\end{aligned}
\tag{10}
$$

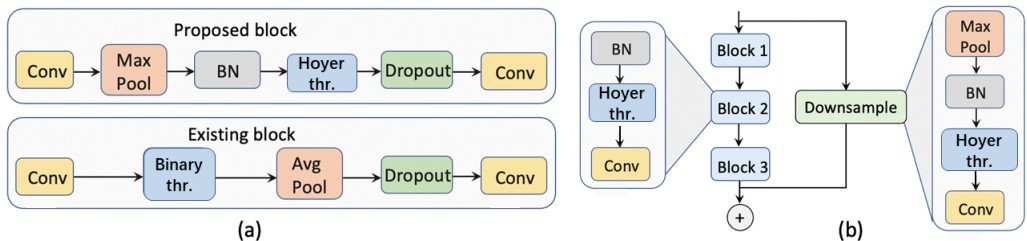

Figure 3: BANN architectures corresponding to (a) VGG and (b) ResNet based models.

This is because $L_H$ is the sum of the Hoyer regularizer of the first $l-1$ layers, the $l$th layer, and the layers after the $l$th layer. While computing $\frac{\partial L_H}{\partial u_l}$, the gradient of the $1^{st}$ part is 0, and the gradient of the $2^{nd}$ ($3^{rd}$) part is captured in the second (first) term in Eq. 8. Note that $\frac{\partial L_H}{\partial u_{l+1}}$ is the gradient backpropagated from the $(l+1)^{th}$ layer, that is iteratively computed from the last layer $L$ (see Eqs. 7 and 11). Also, note that for any hidden layer $l$, there are two gradients that contribute to the Hoyer loss with respect to the activation $u_l$; one is from the subsequent layer $(l+1)$ and the other is directly from its Hoyer regularizer. Similarly, $\frac{\partial L_{CE}}{\partial o_l}$ is computed iteratively, starting from the penultimate layer $(L-1)$ defined in Eq. 7, as follows.

$$\frac{\partial L_{CE}}{\partial o_l} = \frac{\partial L_{CE}}{\partial o_{l+1}} \frac{\partial o_{l+1}}{\partial z_{l+1}} \frac{\partial z_{l+1}}{\partial u_{l+1}} \frac{\partial u_{l+1}}{\partial o_l} = \frac{\partial L_{CE}}{\partial o_{l+1}} \frac{\partial o_{l+1}}{\partial z_{l+1}} \frac{w_{l+1}}{v_l^{th}} \tag{11}$$

Finally, the threshold update for the hidden layer $l$ is computed as

$$\Delta v_l^{th} = \frac{\partial L_{CE}}{\partial v_l^{th}} + \lambda_H \frac{\partial L_H}{\partial v_l^{th}} = \frac{\partial L_{CE}}{\partial o_l} \frac{\partial o_l}{\partial z_l} \frac{\partial z_l}{\partial v_l^{th}} + \lambda_H \frac{\partial L_H}{\partial v_l^{th}}$$
$$= \frac{\partial L_{CE}}{\partial o_l} \frac{\partial o_l}{\partial z_l} \frac{-u_l}{(v_l^{th})^2} + \lambda_H \frac{\partial L_H}{\partial u_{l+1}} \frac{\partial u_{l+1}}{\partial v_l^{th}} \tag{12}$$

$$\frac{\partial u_{l+1}}{\partial v_l^{th}} = \frac{\partial u_{l+1}}{\partial o_l} \cdot \frac{\partial o_l}{\partial v_l^{th}} = w_{l+1} \cdot \frac{\partial o_l}{\partial z_l} \cdot \frac{-u_l}{(v_l^{th})^2} \tag{13}$$

Note that we use this $v_l^{th}$, which is updated in each iteration, to estimate the threshold in our BANN model using Eq. 4.

## A.2  NETWORK ARCHITECTURE

We adopt a series of network architectural tricks for our BANNs (Datta & Beerel, 2022; Chowdhury et al., 2021; Rathi et al., 2020a). As shown in Fig. 3(a), for the VGG variant, we use the max pooling layer immediately after the convolutional layer that is common in many BNN architectures (Rastegari et al., 2016), and introduce the BN layer after max pooling. Similar to recently developed multi-time-step SNN models (Zheng et al., 2021; Li et al., 2021a; Deng et al., 2022; Meng et al., 2022), we observe that BN helps increase the test accuracy with one time step. In contrast, for the ResNet variants, inspired by (Liu et al., 2018b), we observe models with shortcuts that bypass every block can also further improve the performance of the SNN. We also observe that the sequence of BN layer, Hoyer spike layer, and convolution layer outperforms the original bottleneck in ResNet. More details are shown in Fig. 3(b).

## A.3  ABLATION STUDIES

We conduct ablation studies to analyze the contribution of each technique in our proposed approach. For fairness, we train all the ablated models on CIFAR10 dataset for 400 epochs, and use Adam as the optimizer, with 0.0001 as the initial learning rate. Our results are shown in Table 8, where the model without Hoyer threshold layer indicates that we set the threshold as $v_l^{th}$ similar to existing works (Datta & Beerel, 2022; Rathi et al., 2020a) rather than our proposed Hoyer extremum.

Table 8: Ablation study of the different methods in our proposed training framework on CIFAR10.

| Arch. | Hoyer Reg. | Hoyer Spike | Acc. (%) | Sparsity (%) |
|-------|-----------|-------------|----------|--------------|
| VGG16 | × | × | 88.42 | 84.38 |
| VGG16 | ✓ | × | 90.45 | 79.52 |
| VGG16 | × | ✓ | 92.90 | 78.30 |
| VGG16 | ✓ | ✓ | **93.13** | 77.43 |
| ResNet34 | × | × | 87.41 | 77.22 |
| ResNet18 | ✓ | × | 90.95 | 79.50 |
| ResNet18 | × | ✓ | 91.17 | 74.13 |
| ResNet18 | ✓ | ✓ | **91.48** | 74.17 |

Table 9: Comparison of our BANNs with full-precision DNNs & BNNs on VOC2007 dataset.

| Framework | Backbone | mAP(%) |
|-----------|----------|--------|
| Faster R-CNN | Original ResNet50 | 79.5 |
| Faster R-CNN | Bi-Real (Liu et al., 2018b) | 65.7 |
| Faster R-CNN | ReActNet (Liu et al., 2020b) | 73.1 |
| Faster R-CNN | BANN ResNet50 | 73.7 |
| Retinanet | Original ResNet50 | 77.3 |
| Retinanet | BANN ResNet50 (ours) | 70.5 |
| YOLO | BANN DarkNet (Kim et al., 2019) | 53.01 |
| SSD | BNN VGG16 (Wang et al., 2020c) | 66.0 |

With VGG16, our Hoyer regularizer approach leads to a 2.03% increase in accuracy. Together, with our Hoyer threshold layer, the accuracy improves by 2.68% to 93.13% while also yielding a 2.09% reduction in sparsity.

We observe a similar trend for our network modifications and Hoyer threshold layer with ResNet18. However, Hoyer regularizer in this case also increases the sparsity from 72.38% to 77.22% to 79.50%, while also negligibly reducing the accuracy. In summary, the combination of our Hoyer regularizer and Hoyer threshold layer yield the SOTA BNN/SNN performance.

## A.4 OBJECT DETECTION RESULTS

**Object Detection Results**: For object detection on VOC2007, we compare the performance obtained by our BANN models with full-precision DNNs and BNNs in Table 9. For two-stage architectures, such as Faster R-CNN, the mAP of our BANN models surpass the existing BNNs by >0.6%. For one-stage architectures, such as RetinaNet (chosen because of its SOTA performance), our BANN models with a ResNet50 backbone yields a mAP of 70.5% (highest among existing BNN, SNN, AddNNs). Note that our BANN-based VGG and ResNet-based backbones lead to a significant drop in mAP with the YOLO framework that is more compatible with the DarkNet backbone (even existing DarkNet-based SNNs lead to very low mAP with YOLO as shown in Table 9).

## A.5 PROOF OF THRESHOLD DOWNSCALING WITH HOYER EXTREMUM

In order to prove that our Hoyer regularized threshold is always less than or equal to the trainable threshold $v^{th}$, we first prove that the Hoyer extremum of $z_l^{clip}$ is less than or equal to 1. Let us use $c_l$ to represent $z_l^{clip}$, so $\forall j, 0 \le c_l^j \le 1$

$$
\begin{aligned}
Ext(c_l) = \frac{\|c_l\|_2^2}{\|c_l\|_1} &= \frac{\sum_j (c_l^j)^2}{\sum_j c_l^j} \\
&\le \frac{\sum_j (c_l^j \cdot max(c_l^j))}{\sum_j c_l^j} \le max(c_l^j)) \le 1
\end{aligned}
\tag{14}
$$

Table 10: Comparison of our one- and multi-time-step SNN models to existing SNN models on DVS-CIFAR10 and IBM-Gesture datasets.*Results are based on our replicated implementation.

| Dataset | Reference | Training | Architecture | Acc. (%) | T.S. |
|---|---|---|---|---|---|
| DVS-CIFAR10 | (Deng et al., 2022) | TET | VGGSNN | 83.17
75.20* | 10
4* |
| | (Li et al., 2022) | tdBN+NDA | VGG11 | 81.7 | 10 |
| | (Kim & Panda, 2021) | SALT+Switched BN | VGG16 | 67.1 | 20 |
| | This work | Hoyer reg. | VGGSNN | **83.68**
**76.17** | 10
4 |
| N-CalTech101 | (Li et al., 2022) | tdBN+NDA | VGG11 | 78.6 | 10 |
| | (Kim & Panda, 2021) | SALT+Switched BN | VGG16 | 55.0 | 20 |
| | This work | Hoyer reg. | VGG11 | **79.92** | 10 |

So the Hoyer extremum of $z_l^{clip}$ is always less than or equal one, and thus, our Hoyer regularized threshold, which is the product of $v_l^{th}$ and $Ext(z_l^{clip})$ is always less than or equal to the trainable threshold $v_l^{th}$.

## A.6 DATASETS & HYPERPARAMETERS

For training VGG16 models, we using Adam optimizer with initial learning rate of 0.0001, weight decay of 0.0001, dropout of 0.1 and batch size of 128 in CIFAR10 for 600 epochs, and Adam optimizer with weight decay of $5e{-}6$ and with batch size 64 in ImageNet for 180 epochs. For training ResNet models, we using SGD optimizer with initial learning rate of 0.1, weight decay of 0.0001 and batch size of 128 in CIFAR10 for 400 epochs, and Adam optimizer with weight decay of $5e{-}6$ and with batch size 64 in ImageNet for 120 epochs. We divide the learning rate by 5 at 60%, 80%, and 90% of the total number of epochs. Note that $\lambda_h = 1e{-}8$ for all our experiments.

When calculating the Hoyer extremum we implement two versions, one that calculates the Hoyer extremum for the whole batch, while another that calculates it channel-wise. Our experiments show that using the channel-wise version can bring $0.1{-}0.3\%$ increase in accuracy. All the experimental results reported in this paper use this channel-wise version.

For Faster R-CNN, we use SGD optimizer with initial learning rate of 0.01 for 50 epochs, and divide the learning rate by 10 after 25 and 40 epochs each. For Retinanet, we use SGD optimizer with initial learning rate of 0.001 with the same learning rate scheduler as Faster R-CNN.

## A.7 EXTENSION TO SNNs FOR DYNAMIC VISION SENSOR (DVS) TASKS

The inherent temporal dynamics in SNNs may be better leveraged in DVS or event-based tasks (Deng et al., 2022; Li et al., 2022; Kim & Panda, 2021; Kim et al., 2022) compared to standard static vision tasks. Hence, we also evaluate our framework on two DVS datasets, namely DVS-CIFAR10 and IBM-Gesture. As illustrated in Table 10, we surpass the test accuracy of existing works (Li et al., 2022; Kim & Panda, 2021) by 0.93% on average at iso-time-step and architecture. Note that the architecture VGGSNN employed in our work and (Deng et al., 2022) is based on VGG11 with two fully connected layers removed as (Deng et al., 2022) found that additional fully connected layers were unnecessary for neuromorphic datasets. To compensate for the loss of momentum in SGD and improve the accuracy, we adopt the temporal efficient training (TET) method in our framework (Deng et al., 2022). In fact, our accuracy gain is more significant at low time steps, thereby implying the portability of our approach to DVS tasks. Note that similar to static datasets, a large number of time steps increase the temporal overhead in SNNs, resulting in a large memory footprint and spiking activity.

## A.8 ENERGY EVALUATION FRAMEWORK

### A.8.1 COMPUTE ENERGY

The total compute energy (CE) of a BANN ($BANN_{CE}$) can be estimated as

$$BANN_{CE} = DNN_1^{op} E_{mac} + \sum_{l=2}^{L} (S_l DNN_l^{op} E_{ac} + E_{sp} DNN_l^{op}) + \sum_{l=1}^{L} DNN_l^{com} E_{com} \quad (15)$$

because BANN receives full-precision input in the first layer ($l$=1) without any sparsity (Chowdhury et al., 2021; Datta et al., 2022b; Rathi et al., 2020a). Note that $DNN_l^{com}$ denotes the total number of comparison (thresholding) operations in the layer $l$ with each operation consuming 1.64pJ energy in our FPGA platform for FP reperesentation. Also, note that $DNN_l^{op}$ denote the total number of floating point (MAC or AC) operations in layer $l$, where each FP MAC operation consumes 13.32pJ energy, while each FP AC operation consumes 1.8pJ energy. Lastly, $S_l$ denote the spiking activity in the layer $l$, and $E_{sp} = 0.05$pJ denotes the energy overhead due to sparsity, that is incurred in checking whether the binary activation is zero.

The CE of the full-precision DNN ($DNN_{CE}$) and bi-polar BNN ($BNN_{CE}$) is estimated as

$$DNN_{CE} = \sum_{l=1}^{L} DNN_l^{op} E_{mac}, \quad BNN_{CE} = DNN_1^{op} E_{mac} + \sum_{l=2}^{L} DNN_l^{op} E_{xor} \qquad (16)$$

where we ignore the energy consumed by the ReLU operation (significantly lower compared to thresholding operation) since that includes only checking the sign bit. Note that $E_{xor}$ denotes the XOR energy incurred in the binary convolution/linear operations in bi-polar BNNs.

The compute energy of an SNN with a total of $T$ time steps is denoted as

$$SNN_{CE} = \sum_{t=1}^{T} \left( DNN_1^{op} E_{mac} + \sum_{l=2}^{L} (S_l^t DNN_l^{op} E_{ac} + E_{sp} DNN_l^{op}) + \sum_{l=1}^{L} DNN_l^{com} E_{com} \right)$$
$$(17)$$

where $S_l^t$ denotes the activation sparsity of layer $l$ at time step $t$, and we replicate the BANN energy for all $T$ time steps. Note that we ignore the potential reset energy since the number of reset operations is negligible compared to the FLOPs, and the energy consumed per reset operation is similar to that of an AC operation.

Intuitively, BANNs with low-bit ($\leq$6-bits) weights, consume lower compute energy compared to BNNs since the energy savings due to the activation sparsity even with its overhead outweighs the energy savings due to XOR operations compared to low-precision AC operations. BANNs also consume lower compute energy compared to SNNs and DNNs due to the higher activation sparsity, leading to less number of FP operations.

### A.8.2 MEMORY ENERGY

Since memory accesses often dominate the total energy consumption in DNNs, we incorporate the same in our energy model using (Ottati et al., 2023).

The memory energy of a BANN is estimated as

$$BANN_{ME} = \sum_{l=1}^{L} C_l^i C_l^o k_l^2 E_{rd}^{wt} + C_1^i C_1^o k_l^2 H_1 W_1 E_{rd}^{inp} +$$
$$\sum_{l=2}^{L} C_l^i C_l^o k_l^2 H_l W_l (S_l E_{rd}^{sp} + E_{sp}) + \sum_{l=1}^{L} H_l W_l C_l^o E_{wr}^{sp} \qquad (18)$$

where $C_l^i$, $C_l^o$, $k_l$, $H_l$, and $W_l$ denote the number of input channels, number of output channels, kernel size, output feature map height, and output feature map width respectively. Note that $E_{rd}^{sp}$ and $E_{wr}^{sp}$ denote the energy incurred in reading a spike input and writing a spike output respectively from/to the on-chip FPGA memory. On the other hand, $E_{rd}^{inp}$ denotes the read energy of each 8-bit input pixel value, and $E_{rd}^{wt}$ denotes the read energy of the weights from the memory.

The memory energy of a traditional bi-polar BNN is estimated as

$$BNN_{ME} = \sum_{l=1}^{L} C_l^i C_l^o k_l^2 E_{rd}^{wt} + C_1^i C_1^o k_l^2 H_1 W_1 E_{rd}^{inp} + \sum_{l=1}^{L} C_l^i C_l^o k_l^2 H_l W_l E_{rd}^{sp} + \sum_{l=1}^{L} H_l W_l C_l^o E_{wr}^{of}$$
$$(19)$$

where $E_{wr}^{of}$ denotes the energy incurred in writing the FP output feature map to the on-chip memory. While $E_{rd}^{wt}$ is lower for BNNs compared to multi-bit-weighted BANNs, BNNs incur higher output

Table 11: Comparison of the different compute and memory energy numbers for FP weights and membrane potentials obtained from our post place-and-route FPGA simulations on 28nm Kintex7 device.

| Parameter | Energy (pJ) | Parameter | Energy (pJ) |
|---|---|---|---|
| $E_{mac}$ | 13.2 | $E_{rd}^{wt}{=}E_{rd}^{mem}$ | 33.4 |
| $E_{ac}$ | 1.8 | $E_{rd}^{inp}$ | 8.6 |
| $E_{com}$ | 1.4 | $E_{rd}^{sp}$ | 1.7 |
| $E_{sp}$ | 0.05 | $E_{wr}^{sp}$ | 3.8 |
| $E_{xor}$ | 0.08 | $E_{wr}^{mem}$ | 40.1 |

feature map write energy due to multi-bit outputs, and higher input feature map read energy due to absence of spike sparsity.

In summary, BANN can hep reduce both the compute and memory energy compared to BNNs, SNNs, and non-spiking DNNs, provided the hardware supports the activation sparsity.

The memory energy of an SNN with $T$ time steps is estimated as

$$SNN_{ME}{=}\sum_{l=1}^{L}C_l^i C_l^o k_l^2 E_{rd}^{wt}{+}C_1^i C_1^o k_l^2 H_1 W_1 E_{rd}^{inp}{+}$$
$$\sum_{t=1}^{T}\left(\sum_{l=2}^{L}C_l^i C_l^o k_l^2 H_l W_l (S_l^t E_{rd}^{sp}{+}E_{sp}){+}\sum_{l=1}^{L}H_l W_l C_l^o (E_{wr}^{sp} + E_{rd}^{mem} + E_{wr}^{mem})\right) \tag{20}$$

where $E_{rd}^{mem}$ and $E_{wr}^{mem}$ denote the energies incurred in read and write operations of the membrane potential respectively, that happens for every time step. Lastly, the memory energy of a FP DNN is computed as

$$DNN_{ME}{=}\sum_{l=1}^{L}C_l^i C_l^o k_l^2 E_{rd}^{wt}{+}C_1^i C_1^o k_l^2 H_1 W_1 E_{rd}^{inp}{+}\sum_{l=1}^{L}C_l^i C_l^o k_l^2 H_l W_l E_{rd}^{if}{+}\sum_{l=1}^{L}H_l W_l C_l^o E_{wr}^{of} \tag{21}$$

where $E_{rd}^{if}$ denotes the read energy of the full-precision input feature map, that is higher than the spike read energy incurred in the BNN/BANN/SNN. Note that all the memory access energy numbers, $E_{rd}^{wt}, E_{rd}^{sp}, E_{wr}^{sp}, E_{rd}^{inp}, E_{rd}^{if}, E_{wr}^{of}, E_{rd}^{mem}, E_{wr}^{mem}$, are obtained from our FPGA simulation setup with a 42MB of on-chip (BRAM and URAM) capacity. Note that each of these numbers depend on the bit-precision of the value read/written. For example, assuming that 32 spikes are encoded to an 32b memory word, the energy associated with a spike read memory operation is almost same as $\frac{E_{rd}^{wt}}{32}$, for 32-bit FP weights. All these numbers, along with those for the compute operations, are tabulated in Table 11, where the weights, membrane potentials, input and output feature maps have FP representation.

Note that for SNNs, the memory footprint is primarily dominated by the read and write accesses of the post-synaptic potential at each time step (Yin et al., 2022). This is because these memory accesses are not influenced by the SNN sparsity since each post-synaptic potential is the sum of $k^2 c_{in}$ weight-modulated spikes. For a typical convolutional layer, $k = 3$, $c_{in} = 128$, and so it is almost impossible that all the $k^2 c_{in}$ spike values are zero for the membrane potential to be kept unchanged at a particular time step. Note that the number of input (output) feature map read (write) accesses can be reduced with the spike sparsity, and typically do not dominate the memory footprint of SNNs. Since BANNs do not have the temporal membrane potential, they can significantly reduce the memory cost compared to SNNs. Compared to DNNs and BNNs, the memory cost, that is dominated by the feature map accesses (with weight re-use), can be reduced in BANNs by the zero gating logic that leverages spike sparsity.

## A.9 Training & Inference Time Requirements

Because SOTA SNNs require iteration over multiple time steps and storage of the membrane potentials for each neuron, their training and inference time can be substantially higher than the DNN/BNN counterparts. Though reducing their latency

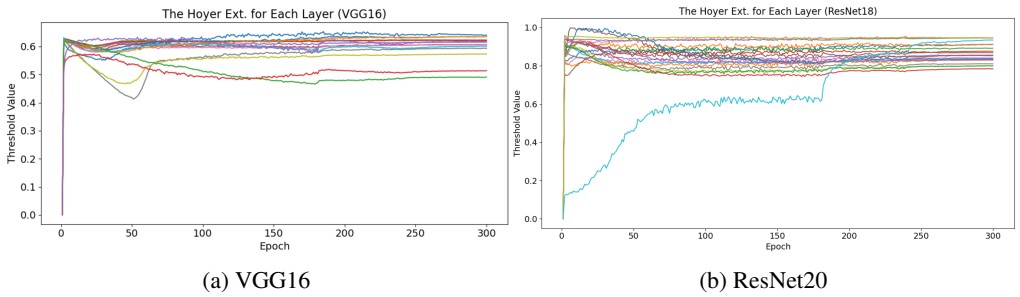

(a) VGG16          (b) ResNet20

Figure 5: Evolvement of the learned threshold value for each layer of VGG16 and ResNet20 on CIFAR10.

to 1 time step can bridge this gap significantly, as shown in Figure 4, it significantly increases the training complexity due to the iterative time-step reduction. On average, our BANNs represent a $2.38\times$ and $2.33\times$ reduction in training and inference time per epoch respectively, compared to the multi-time-step training approaches (Datta & Beerel, 2022; Rathi et al., 2020a) with iso-batch and hardware conditions.

Compared to the existing one-time-step SNNs (Chowdhury et al., 2021), we yield a $19\times$ and $1.25\times$ reduction in training and inference time. This inference time reduction is possibly due to more efficient PyTorch tensor operations used in our code which may be better optimized using the underlying CUDA compiler. Such significant savings in training time, which translates to power savings in big data centers, can potentially reduce AI's environmental impact.

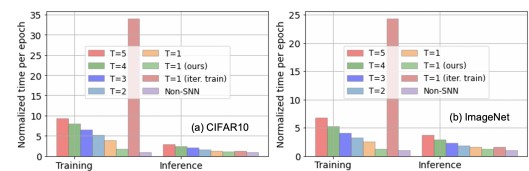

Figure 4: Normalized training and inference time/epoch with iso-batch (256) and hardware (RTX 3090, 24 GB memory) conditions for (a) CIFAR10 and (b) ImageNet with VGG16.

### A.10 COMPARISON OF LATENCY FOOTPRINT

Using a hardware validated latency model on ResNet-18, our 6-bit weighted BANNs incur $1.34\times$ higher latency compared to uni-polar BNNs, as activation sparsity favors energy not latency & popcount incurs lower latency than accumulates. On the contrary, our binary weighted BANNs incur almost similar ($0.96\times$) latency as a uni-polar BNN.

### A.11 STABILITY OF HOYER REGULARIZED TRAINING

The effect of the Hoyer spike layer is a down-scaling of the thresholds obtained from SGD. However, because these thresholds are fixed during inference, the mapping to neuromorphic hardware is no more complex than a traditional SNN. Moreover, since our training method incorporates the learnable training threshold obtained via SGD, we report its evolvement during the training stage in Fig. 5 (a) and (b) for VGG16 and ResNet20 respectively on CIFAR10. As we can see, our threshold value stabilizes well during the later parts of the training for all the layers, demonstrating the robustness of our method.

