# OpenReview forum: "Can we get the best of both Binary Neural Networks and Spiking Neural Networks for Efficient Computer Vision?"
_ICLR.cc/2024/Conference — ICLR 2024 poster_

### Official Review · Reviewer_J4aA · 2023-10-30

**Soundness:** 3 good
**Presentation:** 2 fair
**Contribution:** 2 fair
**Rating:** 5
**Confidence:** 4

**Summary:**

This paper proposed a spiking neural network-based training scheme that eliminates the iterative time step down to 1 while keeping the uni-polar activation scheme (0 and 1). The proposed algorithm is mainly constructed based on a layer-wise learnable threshold trained by the Hoyer regularization. The proposed method achieves comparable accuracy as the prior BNN work on both the CIFAR and ImageNet datasets.

**Strengths:**

The major contribution of the proposed method is reducing the multi-step, iterative spiking process down to one. I agree with the argument that increasing the number of 1s in the output spike can improve the activity of the training process (weight update) and convergence.

Regarding the performance of the proposed method, reporting the accuracy of the classification and object detection datasets is acceptable.

It is helpful that the author can provide some theoretical insights and proofs.

**Weaknesses:**

**W1:** Most of the binary neural network research binarizes the weights weights as well. However, this paper does not report the weight precision and compares the precision scheme with BNN in a comprehensive manner.

**W2:** The performance comparison against SNN is not comprehensive enough. E.g., The recent TET [R1], DSpike [R2], and DSR [R3] are excluded from Table 2.

**W3:** The proposed method shows worse performance on ResNet-50 than VGG-16, which is controversial with the full-precision model performance, why is that?

**W4:** The proposed method incorporates the learnable potential threshold, I think it is important to report the evolvement of the threshold value during the training process, especially the training stability of the threshold value.

**W5:** Although the static image-based classification datasets (e.g., ImageNet) are good indicators of model performance, it is also important to evaluate the performance of SNN with DVS datasets due to the property of spatial-temporal events (e.g., DVS-CIFAR10, N-CalTech101, IBM-Gesture).

[R1]: Temporal Efficient Training of Spiking Neural Network via Gradient Re-weighting

[R2]: Differentiable Spike: Rethinking Gradient-Descent for Training Spiking Neural Networks, NeurIPS'21

[R3]: Training High-Performance Low-Latency Spiking Neural Networks by Differentiation on Spike Representation CVPR 2022

**Questions:**

Please refer to Weaknesses.

---

> ### Author Response · Authors · 2023-11-19
> **Response to Reviewer J4aA**
>
> Thanks for your insightful review and suggestions to improve the quality of our work. Please find our response below.
>
> **BNN weight precision and comprehensive comparison**
>
> The accuracy numbers of our BANNs reported in Table 1 and 2 are with 32-bit floating point weights. We have also shown the BANN accuracies with 1-bit weights in Table 5 for direct comparison with BNNs. As we can see, our BANNs with 1-bit weights yield superior test accuracy compared to existing uni-polar BNNs. However, they do not surpass the test accuracies of state-of-the-art (SOTA) bi-polar BNNs due to the expressivity bottleneck explained in Appendix A.11. We have also shown the accuracies obtained by BANNs for different weight precisions from binary to floating point in Table 4. As we can see, the accuracy decreases progressively with the reduction in bit-precision which is expected.
>
> **Comprehensive performance comparison against SOTA SNNs**
>
> We have now added the comparison of our proposed method with [1-3] as you suggested in Table 2. We have also compared with a few other recently proposed training methods [4-6] in Table 2. Our approach yields superior trade-off between accuracy and number of time steps for most of these works. Our approach at one time step surpasses the reported test accuracy of some of these techniques at more than one time step; other techniques, that are open-sourced, when extended to one time step, yield accuracies that are significantly lower compared to us on CIFAR10 as shown below. (This is based on the replicated implementation in our setup since the result for one time step was not reported).
>
> | **Work** | **Architecture** | **Accuracy** | **Time steps** |
> |------------------|----------------|--------------|------------------|
> | SurrModu  [6]      | ResNet18               |87.02       | 1           |
> | TET [1]           | ResNet19              | 87.84     | 1         |
> | Our work          | ResNet18             | 92.50     | 1        |
>
> **Lower performance of spiking ResNet50 over spiking VGG16**
>
> This phenomenon of ResNets yielding lower performance compared to VGGs is primarily due to the issue of vanishing/exploding gradients in deep ResNets, which was somewhat addressed in the spike-element-wise (SEW) ResNet work [7]. We have now adopted the SEW ResNet architecture to train our SNNs, which yields higher test accuracy in ResNet50 compared to VGG16 on CIFAR10, as shown in the updated Table 1 in the revision. On ImageNet, although we can improve the accuracy of spiking ResNet50 with the SEW architecture, it is still ~1% below compared to VGG16 for one time step. This is consistent with recently published works [3, 8], that observe >3% drop in test accuracy with ResNet34 compared to VGG16 on ImageNet. Note that ResNet50 typically provides <1% accuracy increase from ResNet34 for low-time-step SNNs as shown in [7] and also observed in our experiments. Hence, ResNet50 is expected to yield lower accuracy than VGG16 for the current SNNs. Improving the accuracy of directly trained spiking Resnet models for ultra-low time steps is thus an important research direction.
>
> **Evolvement of threshold during training**
>
> The threshold value stabilizes fairly well during the later parts of the training. As mentioned in the last paragraph of Section 3.1, the threshold, i.e., the Hoyer extremum in each layer changes only slightly during the later stages of training, which indicates that it is most likely an inherent attribute of the dataset and model architecture. We have also added Appendix A.12, along with Fig. 5 to report this phenomenon.
>
> **SNN Performance on DVS datasets**
>
> We had included the performance of our SNNs on DVS-CIFAR10 in Appendix A.6 (now moved to Appendix A.7) of the original manuscript. In the revision, we have also added the performance comparison on another DVS dataset, N-Caltech 101, as you suggested. As shown in Table 10 and below, our BANNs, when extended to multiple time steps for both these tasks, yield $0.9$% higher test accuracy compared to the SOTA TET [1] training approach. This highlights the portability of our approach to DVS tasks. * denotes results based on our replicated implementation.
>
>
> | **Dataset** | **Ref.** | **Architecture** |**Accuracy** | **Time steps** |
> |------------------|----------------|--------------|------------------|------------------|
> | DVS-CIFAR10   | TET [1]              |VGGSNN       | 83.17           | 10      |
> | DVS-CIFAR10   | TET [1]              |VGGSNN       | 75.20*           | 4*    |
> | DVS-CIFAR10   | tdBN-NDA [9]              | VGG11      | 81.7         | 10      |
> | DVS-CIFAR10   | Our work          |VGGSNN       | 83.68          | 10      |
> | DVS-CIFAR10   | Our work            |VGGSNN       | 76.17           | 4   |
> | N-CalTech101  | tdBN-NDA [9]            | VGG11      |78.6        | 10      |
> | N-CalTech101  | Our work          |VGG11       | 79.92           | 10    |

---

> ### Author Response · Authors · 2023-11-19
> **References**
>
> [1] S. Deng et al., "Temporal Efficient Training of Spiking Neural Network via Gradient Re-weighting", ICLR 2022
>
> [2] Y. Li et al., "Differentiable Spike: Rethinking Gradient-Descent for Training Spiking Neural Networks", NeurIPS 2021
>
> [3] Q. Meng et al., Training High-Performance Low-Latency Spiking Neural Networks by Differentiation on Spike Representation CVPR 2022
>
> [4] M. Xiao et al., "On-line training through time for spiking neural networks", NeurIPS 2022
>
> [5] C. Duan et al., "Temporal effective batch normalization in spiking neural network", NeurIPS 2022
>
> [6] S. Deng et al., "Surrogate module learning: Reduce the gradient error accumulation in training spiking neural networks", ICML 2023
>
> [7] W. Fang et al., "Deep Residual Learning in Spiking Neural Networks", NeurIPS 2021
>
> [8] Y. Guo et al., "IM-Loss: Information Maximization Loss for Spiking Neural Networks", NeurIPS 2022
>
> [9] Y. Li et al., "Neuromorphic data augmentation for training spiking neural networks", ECCV 2022

---

> ### Author Response · Authors · 2023-11-22
> **Last day of discussion period**
>
> Dear Reviewer J4aA,
>
> Thanks again for your review. As the author-reviewer discussion period is about to end (Nov 22, end-of-day AoE time), we kindly request you to take a look at our author response (and the updated version of the paper, which incorporates your suggestions) and let us know if your concerns are addressed or if you have any follow-up questions.
>
> Thanks.

---

> > ### Comment · Reviewer_J4aA · 2023-11-22
> > **Feedback from reviewer**
> >
> > Sorry for the late reply, and I appreciate the hard work from the author.
> >
> > Most of my comments were addressed, and I decided to raise my score to 5 (or 5.5).
> >
> > First, it is great to see the improved accuracy of the proposed method with T=1 on DVS datasets.
> >
> > The reason that I don't think this work is an acceptance-ready paper is the insufficient performance (improvements) of the proposed method. Although we can see the increased accuracy by applying the proposed Hoyer regularized to BNN, it seems like the SNN itself still lags behind both BNN and SNN. I totally agree with the author that the update of membrane potential will increase the memory w/r cost, latency, and energy, but it is also true that the full-precision membrane potential can be quantized to low-precision representation, which is relatively more efficient compared to the floating point representation. With the accuracy-driven algorithm design, it might be okay for people to trade in extra memory costs for better accuracy.

---

> ### Author Response · Authors · 2023-11-23
> **Response to reviewer**
>
> Thanks for your kind response, and deciding to raise your score. We would like to clarify a few things here; please see your response below.
>
> **Comparison with BNN**: We have demonstrated that our BANNs can surpass the accuracy of state-of-the-art (SOTA) uni-polar BNNs as shown in Table 5. In case you have missed, we have also shown that our Hoyer regularized training, when applied to state-of-the-art (SOTA) bi-polar BNNs, results in  $0.2{−}1.3$% increase in test accuracy in Table 6 and below.
>
> | **Work** | **Accuracy (%)** | **Accuracy w/ Hoyer (%)** |
> |------------------|----------------|--------------|
> | PokeBNN-1.0x   | 73.4            | 73.6 (+0.2)    |
> | MeliusNet59   | 71.0        | 72.3 (+1.3)  |
> | ReCU (ResNet18)   | 66.4            | 67.8 (+1.4)   |
>
> This is done with the same way as using the Hoyer extremum of the normalized and clipped activation as the threshold, where the clipping is done at -1 and +1 (instead of 0 and +1 for our BANNs). The equations below govern the operation of our Hoyer regularized bi-polar BNN.
>
>
>  $\boldsymbol{z}_l^{clip}{=} 1, \ \  \text{if }  \boldsymbol{z}_l{>}1 $
>
> $\ \ \ \  \ \   =\boldsymbol{z}_l, \ \ \text{if } -1{\leq}\boldsymbol{z}_l{\leq}1 $
>
> $\ \ \ \  \ \   =0, \ \ \text{if } \boldsymbol{z}_l < 0$
>
>    $\boldsymbol{o}_l{=} h_s(\boldsymbol{z}_l){=}  1, \text{if } \boldsymbol{z}_l{\geq}E( {\boldsymbol{z}_l^{clip}}) $
>
> $\ \ \ \  \ \   \  \ \  \  \ \ \  \ \ \  \  {=}{-1}, \text{otherwise} $
>
> **Comparison with SNN**: As shown in Table 2, our BANNs surpass the accuracy of most existing SNNs with iso-time-step (T>1). Moreover, we have conducted another ablation study where we evaluated our training approach on the SOTA low-time-step training approach, TET, in an attempt to improve the accuracy. In particular, as shown in TET, we employed the average of the cross-entropy loss with respect to the output spike for every time step (instead of the single cross-entropy loss with respect to the potential at the last time step) with our Hoyer regularized training. As shown below, our approach can indeed improve the SNN test accuracy with low time steps.
>
> | **Dataset** | **Ref.** | **Architecture** |**Accuracy** | **Time steps** |
> |------------------|----------------|--------------|------------------|------------------|
> | CIFAR10   | TET             | ResNet19       | 94.16         | 2      |
> | CIFAR10   | Our work + TET            | ResNet18       | 94.98 (+0.82)          | 2   |
>
> Thus, **even considering the aspect of accuracy-driven algorithm design, we show that our approach can outperform both BNN at iso-expressivity, and SNN at iso-time-step**.
>
> **Memory cost with low-precision membrane potential**
>
> We absolutely agree with you that the membrane potential can be quantized to low-precision representation. Even assuming we can quantize the potential down to 4 bits (though at this point there might be some drop in accuracy), a 2-time-step SNN would incur $2.92\times$ higher memory energy and $1.71\times$ higher compute energy (and $4.5\times$ higher total energy) compared to our 1-time-step BANN, as per our energy model in A.9. For example, let us compare an SNN trained with TET for T=2 and our BANN for T=1. While the SNN trained with TET yields 94.16% accuracy with ResNet19, our BANN yields 93.88% accuracy with a less expressive network, ResNet18 (TET did not report results on the standard ResNet18 architecture). It might not be a wise choice to choose a model that incurs ${>}4.5\times$ higher energy (${>}$ because ResNet19 would incur higher energy compared to ResNet18), while yielding ${<}0.28$% accuracy increase (${<}$ because 4-bit potential would result in accuracy drop compared to FP), especially in resource-constrained devices. Even if it is possible to trade-in these extra energy costs for better accuracy, **our approach can also scale to multiple time steps ($T{>}1$) as shown below and surpass the accuracy of existing training methods**.
>
> | **Dataset** | **Ref.** | **Architecture** |**Accuracy** | **Time steps** |
> |------------------|----------------|--------------|------------------|------------------|
> | CIFAR10   | TET         | ResNet19       |  94.16         | 2      |
> | CIFAR10   | Our work             | ResNet18       | 93.88           | 1    |
> | CIFAR10   | Our work            | ResNet18     | 94.62         | 2      |
>
> Our approach can thus enable a wide range of SNNs (with different values of T) with different accuracy-energy trade-offs that can be deployed with diverse resource budgets.
>
> We hope this response can convince you about the superiority of our training method in the context of both BNNs and SNNs compared to existing works.

---

### Official Review · Reviewer_X4qq · 2023-10-31

**Soundness:** 3 good
**Presentation:** 2 fair
**Contribution:** 3 good
**Rating:** 6
**Confidence:** 4

**Summary:**

The presented method trains binary neural networks with the goal of sparse activations. The is achieved by borrowing concept from SNN training that use a threshold function in the neuron model to generate output spikes. In this method, the threshold value is determined for each layer using the Hoyer extremum. The method was tested on CIFAR-10 and ImageNet datasets and achieved superior accuracy and energy efficiency.

**Strengths:**

BNNs can certainly benefit from sparsity, just as SNNs do. Hence, the approach to transfer the SNN thresholding concept to BNNs with binary activations 0 and 1 is promising. With the right hardware, operations can be saved. The effect of sparsity becomes clear when compared with BNNs that use binary weights. The proposed method seems more energy efficient despite using floating-point weights.
The activations achieve a high sparsity. And by using the Hoyer regularisation during training, activations are resistant to noise and achieve higher accuracy than most SNNs, especially ones with only 1 time step.

**Weaknesses:**

The experiments executed in this paper are generally insightful. It is especially interesting to compare with both BNNs and SNNs. However, some evaluations are lacking detail. The energy consumption is estimated using power numbers of a paper from 2014 at an outdated technology node. Real values obtained from logic synthesis will be significantly different. Additionally, sparse operation induces overhead, such as checking if an activation is actually zero. Those factors are not accounted for in the estimations. Since there was an FPGA simulation done to evaluate the effect of quantisation, I would like to see a power estimation using this FPGA simulator. Alternatively, a GPU evaluation of the actual power consumption is interesting, since it is stated that the BNN operations are compatible with “existing hardware (standard GPUs)”.

For comparison of the energy consumption between SNNs and the proposed method, sparsity is used as a proxy. However, it is not clear how the sparsity numbers in Figure 3 were obtained except that they “represent existing low-latency SNN works”. Were those numbers given by the references or was it a replicated implementation by the authors?

More emphasis could be placed on the comparison with BNNs. It is stated that SOTA BNNs use network modifications that increase FLOPS but also expressivity. To which extend does calculating the Hoyer extremum during inference incur additional FLOPS? The power estimation in Equation 9 only accounts for one additional comparison.

Important parameters such as λ are not provided. At the same time, it was not indicated that training code and trained models will be provided. That means it is currently difficult to validate the results of this paper and derive further research. I would see open sourcing as a requirement of acceptance.

The formatting of the references in the text makes the paper very hard and slow to read. Figure 2 was prominently placed in the manuscript but never mentioned or further explained in the text.

**Questions:**

During inference, the Hoyer extremum is used to do the binarisation of the activations. How does is affect the number of FLOPS in Equation 9? How can the Hoyer extremum be reduced to a simple comparison to make the inference match the equation?

Could you take some actual energy measurements using your already existing FPGA simulator? It would also be interesting to see SNN’s energy consumption on hardware. This way, we do not need to rely on outdated energy numbers from 2014 and also account for the overhead induced by the sparsity.

Why is the ImageNet energy lower than that for CIFAR-10 in Table 4? Could you elaborate how you arrived at those numbers?

---

> ### Author Response · Authors · 2023-11-19
> **Response to Reviewer X4qq**
>
> Thank you very much for your detailed review and suggestions to improve the quality of our work. Please find our response below.
>
> **Updated power estimation using FPGA simulations**
>
> We have conducted FPGA simulations to extract the energy incurred by the accumulate and threshold comparison operation of the SNN and multiply-and-accumulate operation of the DNN. We have also included the overhead of sparsity (energy required to check if the activation is zero) in our energy evaluation. Additionally, based on the suggestion of Reviewer gkE7, we have also included the memory cost of the weights (and membrane potential for the multi-time-step SNNs) in our evaluation. The details of our improved energy estimation model and FPGA simulation results are provided in the newly created Appendix A.9 in the revision. We have now updated Fig. 3c, Table 4, and Table 5 with energy (per inference) numbers obtained from our improved model. With an FPGA clock frequency of 100 MHz, our BANNs approximately yield a $70.28\times$ and $5.15\times$ decrease in power consumption on average compared to a traditional DNN and a 5-time-step SNN respectively on CIFAR10.
>
> **How is activation sparsity estimated for different time steps?**
>
> The sparsity numbers in Fig. 3 were calculated based on the results of the training framework proposed in reference [1]. Since [1] did not show the sparsity numbers for different time steps, we ran the open-sourced training implementation of [1], and reported the sparsity numbers obtained in our setup for 1-5 time steps. We believe this would lead to a fair comparison because our training method is based on the same surrogate gradient function as [1]. These sparsity numbers can certainly change with the training algorithm, however, for any algorithm, the sparsity would increase with the reduction of the number of time steps, because each neuron would get less opportunity to fire spikes.
>
> **Additional FLOPs for calculating Hoyer extremum during inference**
>
> As mentioned in the last paragraph of Section 3 of the paper, the Hoyer extremum in each layer changes only slightly during the later stages of training, which indicates that it is most likely an inherent attribute of the dataset and model architecture. Hence, to estimate the threshold, we calculate the exponential average of the Hoyer extremums during training, similar to batch normalization (BN) layers, and use this estimated value during inference. Thus, the threshold remains fixed during inference, and does not incur any additional FLOPs.
>
> **Value of lambda and open-sourcing code**
>
> We use lambda=1e-8 for all our experiments, which is now reported in Appendix A.6. We are currently cleaning up our code and will attach our training code and logs as soon as possible.
>
> **Formatting of references and confusion with Fig. 2**
>
> We sincerely apologize for the mis-formating of the references and obscuring your reading of our paper. We have now fixed the issue in the revision. We also apologize for missing the description of Fig. 2 in the main text. We have now moved the figure, along with its description in Appendix A.2.
>
> **Lower energy numbers for ImageNet compared to CIFAR10**
>
> We apologize for a mistake made here; we had unintentionally interchanged the energy numbers of CIFAR10 with ImageNet in Table 4. We have now corrected the mistake in the revision. Additionally, we have now incorporated the overhead of sparsity and the memory access costs in the energy numbers shown in Table 4 and Table 5. All these energy numbers are obtained from the model illustrated in Appendix A.9.
>
> [1] Rathi et al., "DIET-SNN: Direct Input Encoding With Leakage and Threshold Optimization in Deep Spiking Neural Networks", TNNLS 2021

---

> > ### Author Response · Authors · 2023-11-21
> > **Open-sourcing our code**
> >
> > Dear Reviewer X4qq,
> >
> > We have now uploaded our code to 'Supplementary material'. We sincerely hope that our response has addressed your concerns. We would be happy to answer any additional questions that you may have.
> >
> > Best,
> >
> > Authors

---

> > ### Comment · Reviewer_X4qq · 2023-11-22
> >
> > Thanks to the authors for the clarification. It resolves some of my concerns, so I raised my rating.

---

> > > ### Author Response · Authors · 2023-11-22
> > > **Thanks for raising your rating!**
> > >
> > > Dear Reviewer X4qq,
> > >
> > > We are delighted to know that our response addressed some of your concerns. We would also like to thank you for increasing your rating. Please let us know if you need any additional clarifications.
> > >
> > > Best,
> > >
> > > Authors

---

> ### Author Response · Authors · 2023-11-22
> **Last day of discussion period**
>
> Dear Reviewer X4qq,
>
> Thanks again for your review. As the author-reviewer discussion period is about to end (Nov 22, end-of-day AoE time), we kindly request you to take a look at our author response (and the updated version of the paper, which incorporates your suggestions) and let us know if your concerns are addressed or if you have any follow-up questions.
>
> Thanks.

---

### Official Review · Reviewer_gkE7 · 2023-10-31

**Soundness:** 3 good
**Presentation:** 3 good
**Contribution:** 3 good
**Rating:** 6
**Confidence:** 5

**Summary:**

This paper proposes a new training method for binary activation neural networks. This method employs the Hoyer regularizer to each BANN layer to shift the activation values away from the threshold. As a result, the proposed BANN model can achieve better accuracy than other methods.

**Strengths:**

* The proposed method can improve the accuracy of both BANN and SNN models
* Experiments show the energy efficiency comparisons to demonstrate the effectiveness of this method

**Weaknesses:**

* The proposed method is not closely related to SNN, and it mainly focuses on the BANN. This method can be used in SNNs for better model accuracy, but it can not bridge the gap between BNN and SNN as the title said. It would be preferable to discuss more about the relationship between BNN, SNN, and the proposed method to illustrate how this method bridges BNN and SNN.
* The energy evaluation method in the experiments is inaccurate. Usually, memory accesses take a large proportion of energy consumption. However, in the experiments, it only estimates the computed energy. It is better to consider the energy of memory access as well.

**Questions:**

* In section 2.2, “Note that this model is similar to leaky-integrate-and-fire (LIF) model”. This is not very direct. Why they are similar? In this model, there is no membrane potential, which is important in LIF.
* Typo: in section 4, Accuracy Comparison with SNNs, “Table 7” should be “Table 2”

---

> ### Author Response · Authors · 2023-11-19
> **Response to Reviewer gkE7 [1]**
>
> Thanks a lot for your review and valuable suggestions to improve the quality of our work. Please find our response below.
>
> **Relationship between BNN and SNN**
>
> Our BANNs are identical to one-time-step SNNs, and can be readily extended to traditional multi-time-step SNNs with the incorporation of the membrane potential (state variable). The threshold value of our BANNs remain fixed during inference, as explained in the last paragraph of Section 3.1, and can be used for all time steps in SNNs. This has been illustrated in the newly created Section 5, and added below.
>
> To extend BANNs to SNNs, we **use the traditional LIF model** for the neurons where **the membrane potential integrates the weight modulated input spikes and leaks over time**.
>
> We use the **soft reset mechanism that reduces the membrane potential by the threshold value** when an output spike is generated. It has been shown that soft reset minimizes the information loss by allowing the spiking neuron to carry forward the surplus potential above the firing threshold to the subsequent time step [1,2].
>
> **We use our proposed combination of Hoyer regularized training and Hoyer spike layer to train the per layer threshold, while we train the weights and leak term using BPTT**.
>
> The equations governing our LIF model for our multi-time-step SNNs on static and DVS datasets are shown below.
>
> $U_i^{temp}(t)=\lambda U_i(t-1)+\sum_j W_{ij}{S_j(t)}$
> $S_i(t) = 1 \ \text{if } U_i^{temp}(t)>V^{th} \ \text{else} \ 0$
>
> $U_i(t) = U_i^{temp}(t)-S_i(t)V^{th}$
>
> Here $U_i(t)$ denotes the membrane potential of the $i^{th}$ neuron at time step $t$, $S_i(t)$ denotes the binary output of the $i^{th}$ neuron at time step $t$, $V^{th}$ denotes the threshold, $\lambda$ denotes the leak, and $W_{ij}$ denotes the weight connecting the pre-synaptic neuron $j$ and the neuron $i$. Our BANNs, when extended to SNNs with multiple time steps, can lead to a small but significant increase in test accuracy as shown in Table 6, thereby demonstrating the efficacy of our approach. From another perspective, it can be argued removing the temporal dimension from the LIF model above leads to a BANN.
>
> Our BANNs are also identical to previously proposed uni-polar sparse BNNs [3, 4] when the weight precision is quantized down to 1-bit. Moreover, we show that our training method results in our 1-bit-weighted BANNs having higher test accuracy compared to existing uni-polar BNNs in Table 5.
>
> **Bridging the gap between BNN and SNN**
>
> The sparse activation in BANNs bears resemblance to SNNs while the difference is that there is no temporal dimension. On the other hand, the 1-bit sparse activation in BANNs bears resemblance to BNNs, in particular uni-polar BNNs, while the difference is that the weight precision is multi-bit. Thus, it can be argued that BANNs bridge the gap between BNNs and SNNs by bringing the activation sparsity of SNNs to BNNs, and removing the temporal dimension of SNNs for equivalence with BNNs. This has been highlighted in the newly added Section 5 of the revision.
>
> That said, we agree that our current title might not reflect our contributions very accurately. We have changed the title to ‘Can we get the best of both SNNs and BNNs for efficient computer vision?’. Please let us know if this title looks good to you.
>
> **Incorporation of memory accesses in energy evaluation**
>
> Our BANNs significantly reduce the memory energy compared to existing multi-time-step SNNs since the latter requires the membrane potentials and weights to be fetched from and read to the on-/off-chip memory for each time step. Our BANNs can avoid these repetitive read/write operations as it does involve any state and lead to a $T\times$ reduction in the number of these accesses compared to a $T$-time-step SNN model.
>
> Compared to traditional bi-polar BNNs, our BANNs can also reduce the number of memory accesses with the support of zero gating logic leveraging the high activation sparsity. This can be achieved by skipping the reading of the spike activation when it is zero.  However, the exact savings in memory energy will depend on the data reuse scheme and the underlying hardware.
>
> We have now incorporated the memory accesses of the weights (and membrane potentials for SNNs) in our energy evaluation framework assuming a simple weight stationary dataflow, as shown in Appendix A.9 in the revision. We perform FPGA simulations to estimate the memory energy incurred in reading an 8-bit weight or membrane potential from the on-chip memory (BRAM and URAM) to the combinational logic. Based on the suggestions of reviewer X4qq, we have also incorporated the overhead of sparsity in our evaluation. We observe that our BANNs yield $70.28\times$ and $5.15\times$ lower energy compared to a traditional DNN and a 5-time-step SNN respectively with our improved energy (compute+memory) evaluation framework. We have updated our energy numbers in Fig. 2, Table 4, and Table 5 obtained from this framework.

---

> > ### Author Response · Authors · 2023-11-19
> > **Response to Reviewer gkE7 [2]**
> >
> > **Inappropriate use of the term ‘LIF model’ and wrong table reference**
> >
> > We apologize for the confusion. In the original version, we stated that our proposed model is similar to LIF without the temporal dimension, where there is no concept of membrane potential. However, we agree this may confuse the readers, and we have now removed this sentence. Lastly, thanks for catching the incorrect table reference. We have now corrected the table reference in the revision.

---

> > ### Comment · Reviewer_gkE7 · 2023-11-23
> >
> > Thanks for the feedback. It is clear to me for the relationship between BNN, SNN, and BANN, and my concerns are mostly addressed, so I raised my rating.

---

> > > ### Author Response · Authors · 2023-11-23
> > > **Thanks for raising your rating!**
> > >
> > > Dear reviewer gkE7,
> > >
> > > Thanks for your insightful review again, that clearly helped to improve our paper. Many thanks for raising your rating. Please let us know if you need additional clarifications.
> > >
> > > Best,
> > >
> > > Authors

---

> ### Author Response · Authors · 2023-11-22
> **Last day of discussion period**
>
> Dear Reviewer gkE7,
>
> Thanks again for your review. As the author-reviewer discussion period is about to end (Nov 22, end-of-day AoE time), we kindly request you to take a look at our response (and the updated version of the paper which incorporates your suggestions) and let us know if your concerns are addressed or if you have any follow-up questions.
>
> Thanks.

---

### Author Response · Authors · 2023-11-21
**Looking forward to further discussions**

Dear Reviewers,

We sincerely hope that our response and the revised manuscript have addressed your concerns. We have highlighted our changes in the revision. In particular, we believe we have clarified the two key concerns regarding energy evaluation and comparisons with other SNNs. We have also uploaded our code in 'Supplementary Material'. Based on your suggestions, we have conducted multiple experiments and added discussions highlighted below. We believe that these experiments and discussions further demonstrate the efficacy of our approach.

1. Incorporation of memory access and overhead of sparsity in the energy evaluation setup, FPGA simulations to capture their energies, and energy models to compare BANNs, BNNs, SNNs, and DNNs (please see Appendix A.9)
2. Comparison of BANNs, extended to SNNs with, with state-of-the-art SNNs on DVS datasets (please see Appendix A.7)
3. Relationship between SNN, BNN, and BANN (please see Section 5)
4. Robustness of our trainable threshold (please see Appendix A.12)

Since the discussion period is approaching towards its end, we would appreciate it if you could let us know whether you have any additional questions or suggestions. We are looking forward to discussions that can further improve our work. Thanks!

Best regards,

The Authors

---

### Meta-Review · Area_Chair_GJAk · 2023-12-14

**Metareview:**

This paper proposes sparse binary activation neural networks (BANNs) as a way to combine the benefits of BNNs (which have good accuracy) and SNNs (which have low energy consumption) for computer vision. They show that their BANNs can get better accuracy and still have low energy consumption. This is an important research area and authors did a great job during the rebuttal to revise the paper with more experiments.

There were several questions raised by the reviewers and authors provided the convincing rebuttal. All the reviewers raised their score.

I recommend an acceptance.

**Justification For Why Not Higher Score:**

It is a solid work. But only a small fraction of ICLR participants will be interested in this work and hence poster.

**Justification For Why Not Lower Score:**

There is no reason to reject this work. The paper makes a clear contribution to the field of sparse/binary neural networks.

---

### Decision · Program_Chairs · 2024-01-16

Accept (poster)